# The Impact of the COVID-19 Lockdown on Quality of Life in Adult Cochlear Implant Users: A Survey Study

Giada Cavallaro *, Alessandra Murri, Emer Nelson, Rosaria Gorrasi and Nicola Quaranta

Otolaryngology Unit, Department of Basic Medical Sciences, Neuroscience and Sense Organs, University of Bari "Aldo Moro", 70010 Bari, Italy
* Correspondence: giadacavallaro@live.it; Tel.: +39-33-93583-495

**Abstract:** Background: The COVID-19 pandemic rapidly spread through Europe in the first months of 2020. On the 9th of March 2020, the Italian government ordered a national lock-down. The study's objectives were: to investigate the effect of lockdown on CI users; and to detect the difference in the perception of discomfort existing between unilateral cochlear implant (UCI) users and bilateral cochlear implant (BCI) users, due to the lockdown experience. Methods: A 17-item, web-based, anonymous online survey was administered to 57 CI users, exploring hearing performance, emotions, practical issues, behavior, and tinnitus. Participation in the study was voluntary. Results: all CI users obtained an abnormal score in all questionnaire themes. For the emotion theme and the practical issue theme, the age range 61–90 showed a significant difference between UCI and BCI users in favor of BCI users (emotion theme: UCI mean = 3.9, BCI mean = 2.3, *p* = 0.0138; practical issues: UCI mean = 4, BCI mean = 3, *p* = 0.0031). Conclusions: CI users experienced the lockdown negatively as regards behavior, emotions, hearing performance, and in practical issues. CI subjects with UCI in old age suffered more from the experience of lockdown than subjects with BCI in the same age, with regards to emotions and practical issues.

**Keywords:** lockdown; cochlear implant; quality of life

## 1. Introduction

The COVID-19 pandemic rapidly spread through Europe in the first months of 2020. Italy was one of the most affected countries in Europe and the second for the number of deaths [1]. On the 9th of March 2020, the Italian Government ordered a national lock-down, prohibiting all unnecessary movement of the population except for work, health conditions, and necessities. The lockdown measures adopted to mitigate the spread of the virus in Italy had paralyzed almost every social and economic aspect of the nation [2]. The World Health Organization (WHO) advised governments to encourage the general public to wear masks in specific situations and environments as part of a comprehensive approach to suppress SARS-CoV-2 transmission [3]. Trecca et al (2020) [4] showed preliminary results on the impact of the use of face masks by medical personnel on the perceived difficulties of 59 adults with hearing loss during their hospital visits. Mild to severe problems were experienced by 86.4% of these patients. The main problem with the face masks was the impossibility of lip reading, and sound attenuation was the main problem for 26 people. Listeners with hearing loss can compensate for reduced audibility by using visual cues (e.g., lip reading) [5]. Unfortunately, the visual cues are largely obscured by the most commonly used masks, including surgical and N95 masks [6]. The attenuation of high-frequency information caused by face coverings demonstrated in previous studies may significantly negatively impact speech recognition even when the listener is using hearing technology, such as a cochlear implant (CI) [7].Cochlear implantation is recommended for patients with severe-to-profound hearing loss [8] and the majority of CI patients experience significant improvements in speech recognition as compared to preoperative performance with hearing

aids, though speech recognition remains poorer than normal-hearing listeners [9,10]. One reason for this discrepancy is the limited spectral information available to CI users, due to a discrete number of electrode contacts and current spread in the cochlea [11]. In our Cochlear Implant Center in Policlinico of Bari, many CI users complained about face masks during their visits to our outpatient clinic because of the negative impact on their communication. Vos et al. [12] demonstrated that CI patients experience significantly impaired recognition of speech produced with an N95 mask and face shield compared to an N95 alone, or no face covering. Naylor et al [13], through a rapidly deployed 24-item online questionnaire, asked subjects with hearing loss about the effects of certain aspects of lockdown, including face masks, social distancing, and video calling, on participants' behavior, emotions, hearing performance, practical issues, and tinnitus, pointing out important issues in subjects with hearing aids. The study's primary aim was to investigate the effect of lockdown on adults that are CI users through an online survey inspired by that of Naylor et. Al [13]; in addition, the secondary objective was to detect the difference in perception of discomfort due to the lockdown experience existing between unilateral cochlear implant (UCI) users and bilateral cochlear implant (BCI) users.

## 2. Materials and Methods

All procedures contributing to this work comply with the ethical standards of the relevant national and institutional guidelines on human experimentation and with the Helsinki Declaration of 1975, as revised in 2008. The study was conducted prospectively. Sensitive data were not collected. Participation in the study was voluntary. The study was approved by the local hospital ethics committee.

### 2.1. Participants

A total of 57 subjects were enrolled, 16 were males (28.1 %) and 41 females (71.9%). The study sample consisted of 23 (40.4%) reported as UCI users, and 34 (59.6%) as BCI users. Participation occurred on a voluntary basis and participants found and filled in the questionnaires by themselves. Table 1 collates all the quantitative results forming the basis for interpretative and statistical evaluation (Table 1).

**Table 1.** Population characteristics.

| Population Characteristics | |
|:---|:---:|
| N population | 57 |
| UCI (N) | 23 |
| BCI (N) | 34 |
| Female (%) | 71.9 |
| Female UCI (N) | 16 |
| Female BCI (N) | 25 |
| UCI users less than 30 y.o.of age (N) | 11 |
| UCI users from 31 to 60 y.o. of age (N) | 8 |
| UCI users from 61 to 90 y.o. of age (N) | 4 |
| 1–5 Years UCI users (N) | 3 |
| More than 5 years UCI users (N) | 20 |

### 2.2. Procedures

A 17-item, anonymous, web-based survey was translated and adapted from the survey created by Naylor et. al. [13], based on anecdotal reports on mass and social media platforms regarding the specific challenges facing people with hearing loss as a result of the lockdown. A cross-cultural adaptation of the original survey was performed using standard techniques [14]. Participants first responded to three questions about their gender,

age, CI ownership (UCI or BCI) followed by 17 COVID-19-related questions. Quantitative responses were on a five-point Likert scale ranging from "strongly agree" to "strongly disagree", plus "not applicable/not sure". The orientation of the questions was randomly varied, so that "agreement" did not always signify "worse" or "better". "Strongly agree" was considered 5 points of score (worse) and "strongly disagree" was considered 1 point of score (better) for the following questions: 1, 2, 3, 4, 6, 7, 9, 10, 12, 13, 14, and 17. "Strongly agree" was considered 1 point of score (better) and "strongly disagree" was considered 5 points of score (worse) for the following questions: 5,8,11,15, and 16. The 17 items were grouped into five themes exploring different aspects of the patient's life during lockdown: hearing performance (questions number 1, 2, 5, 9, 10, 12, 15, and 16); emotion (questions number 3, 6, 7, 11, and 14); practical issues (question number 4); behavior (questions number 8 and 13); and tinnitus (question number 17). The questions represent aspects of response to lockdown, rather than aspects of lockdown itself (e.g., face masks and video calls) since the former is felt to be more illuminating regarding the particular experience of people with hearing loss. One open-ended, free-text question at the end of the survey, asked participants to describe any other positive or negative effects of lockdown that they had experienced. The survey was administered online using the Google Drive platform and posted on the Italian Facebook group "Affrontiamo la Sordità Insieme" (dealing with deafness together) frequented by people with hearing loss. The questionnaire was administrated from the 7th of January to the 30th of March 2021.

*2.3. Data Analysis*

Data was collected and was divided into the different categories: gender, UCI, BCI, age, years of CI use, and tinnitus. The results were recorded by calculating the average of the responses for each question and then calculating the average of the responses for each theme. Standard deviation (SD) was calculated for each theme. The mean of the results of patients with UCI and those with BCI were compared. We used Student's test with a $p = 0.05$ significance level after evaluating the t value in each theme. The response categories "strongly disagree" and "disagree" were finally collapsed into one "disagree" category, and likewise for the two "agree" categories. Each survey item was analyzed individually by calculating the frequency of agreement, disagreement, and neutrality. Free-text responses were explored inductively by mapping them onto themes established by categories of quantitative survey questions and responses.

**3. Results**

The results of the questionnaire showed that the lock-down period influenced the quality of life of CI patients (Table 2). Except for the tinnitus domain, that was minimally influenced by the restriction measures, in all the other domains the mean of the responses was always greater than 3. In particular, mean scores higher than 4 were reported in questions 1, 2, and 3, that referred to the ability to understand people with face-masks and the need for visual reinforcement in CI users. An average score of less than 3 was obtained in questions 13 and 17, showing that lockdown measures did not reduce the use of aids nor increase tinnitus. A further statistical analysis was carried out after dividing the population by gender (male and female), age (sample under the age of 30, sample age range 31–60, sample age range 61–90) and years of CI use (from 1 to 5 years, more than 5 years). Statistical analysis showed no significant difference (at the 0.05 level) between female and male users (both UCI and BCI) for the following domains: behavior, emotions, hearing performance, and tinnitus. A statistically significant difference was found regarding practical issues between females and males. In fact, both UCI and BCI females reported that wearing face masks interfered with wearing CI ($p = 0.0388$ for UCI and $p = 0.0106$ for BCI). With regard to the age range, there was not a statistically significant difference between UCI users and BCI users for the following themes: hearing performance, behavior, and tinnitus. For the emotion theme and the practical issues theme, patients with the age range 61-90 years showed a higher impact on the quality of life. BCI users reported significantly lower scores

in the emotion domain (UCI mean 3.9 vs. BCI mean 2.3, *p* = 0.0138) and practical issues (UCI mean 4 vs. BCI mean 3, *p* = 0.0031). Interestingly wearing bilateral CI interfered less with face masks than wearing UCI. The length of CI use (1–5 years vs. >5 years) was not associated with a statistically significant difference between groups.

**Table 2.** Q: questions; Mean: Mean of the score for each question; SD: Standard deviation.

| Hearing Performance | | |
|---|---|---|
| **Q** | **Mean** | **SD** |
| 1 (mean) Understanding people wearing face masks is harder because the speech is muffled. | 4.3 | 1.02 |
| 2 (mean) Understanding people wearing face masks is harder because I cannot see their mouth moving. | 3.97 | 1.21 |
| 5 (mean) When people speak to me from a safe distance, I can still hear them well enough. | 3.52 | 1.16 |
| 9 (mean) In video calls, I hear worse than if the other person was in the room with me. | 3.20 | 1.38 |
| 10 (mean) In video calls, I hear worse than if I was talking to the person on the telephone. | 2.88 | 1.53 |
| 12 (mean) Subtitles on video calls help. | 3 | 1.53 |
| 15 (mean) Televised updates about COVID-19 are easy for me to follow. | 3.35 | 1.32 |
| 16 (mean) Radio updates about COVID-19 are easy for me to follow. | 2.67 | 1.38 |
| **Behaviors** | | |
| **Q** | **Mean** | **SD** |
| 8 (mean) I use video calls (Facebook, FaceTime, Google, Skype, Zoom, etc.) more often now than I did before lockdown began. | 3.55 | 1.56 |
| 13 (mean) Since lockdown began, I have been wearing my hearing aids less than usual. | 1.55 | 1.02 |
| **Tinnitus** | | |
| **Q** | **Mean** | **SD** |
| 17 (mean) My tinnitus has been worse since lockdown started. | 1.67 | 1.19 |
| **Emotion** | | |
| **Q** | **Mean** | **SD** |
| 3 (mean) I think key workers should be supplied with clear (transparent) face masks. | 4.44 | 0.99 |
| 6 (mean) It is a relief not to be obliged to attend social gatherings where I will not hear well. | 3.11 | 1.60 |
| 7 (mean) The possibility of having to speak to people wearing face masks or from a distance adds to my anxieties about going to public places (e.g., parks, supermarkets). | 3.44 | 1.56 |
| 11 (mean) I enjoy group video calls (involving more than two people). | 3.02 | 1.46 |
| 14 (mean) I am more affected by my hearing loss than usual. | 2.97 | 1.58 |
| **Practical Issue** | | |
| **Q** | **Mean** | **SD** |
| 4 (mean) Wearing a face mask interferes with wearing my cochlear implant (s). | 3.55 | 1.35 |

Table 3 collates all the quantitative results with the percentage for each answer. The proportions and confidence intervals of the variables were calculated and compared using the Chi-square test. A *p*-value < 0.05 was considered to be statistically significant. A statistically significant difference was found in the proportion of responses between UCI and BCI users for the following domains: 2, 3, 6, 7, 8, 9, 10, 11, 12, 13, 14, 15, and 17 (*p*-value < 0.05).

**Table 3.** Q: question; UCI: unilateral cochlear implant users; BCI: bilateral cochlear implant users; n: number of subjects in the sample; "Disagree" is the sum of "disagree" and "strongly disagree" responses; "Agree" is the sum of "agree" and "strongly agree" responses. The results are expressed in percentages. The Chi-square test ($\chi 2$) was used to compare categorical variables (Agree and Disagree) in UCI and BCI users.

| Q | Item Statement | UCI | | | | BCI. | | | | $\chi 2$ | $p$ |
|---|---|---|---|---|---|---|---|---|---|---|---|
| | | n | Disagree (%) | Neutral (%) | Agree (%) | n | Disagree (%) | Neutral (%) | Agree (%) | | |
| 1 | Understanding people wearing face masks is harder because the speech is muffled. | 23 | 8.4 | 13 | 78.6 | 34 | 8.8 | 14.7 | 76.5 | 0.2096 | NS |
| 2 | Understanding people wearing face masks is harder because I cannot see their mouth moving. | 23 | 4.4 | 13 | 82.6 | 34 | 17.6 | 11.8 | 70.6 | 88.5214 | $p < 0.05$ |
| 3 | I think key workers should be supplied with clear (transparent) face masks. | 23 | 12.2 | 18.2 | 69.6 | 34 | 8.8 | 8.8 | 82.4 | 11.2093 | $p < 0.05$ |
| 4 | Wearing a face mask interferes with wearing my cochlear implant(s). | 23 | 17.4 | 26.1 | 56.5 | 34 | 20.6 | 23.5 | 55.9 | 2.278 | NS |
| 5 | When people speak to me from a safe distance, I can still hear them well enough. | 23 | 26.1 | 21.7 | 52.2 | 34 | 17.7 | 38.2 | 44.1 | 3.5392 | NS |
| 6 | It is a relief not to be obliged to attend social gatherings where I will not hear well. | 23 | 17.4 | 26.1 | 56.5 | 34 | 38.2 | 14.7 | 47.1 | 78.5815 | $p < 0.05$ |
| 7 | The possibility of having to speak to people wearing face masks or from a distance adds to my anxieties about going to public places (e.g., parks, supermarkets). | 23 | 17.4 | 30.4 | 52.2 | 34 | 29.4 | 11.8 | 58.8 | 12.9495 | $p < 0.05$ |
| 8 | I use video calls (Facebook, FaceTime, Google, Skype, Zoom, etc.) more often now than I did before lockdown began. | 23 | 34.8 | 13 | 52.2 | 34 | 23.5 | 17.7 | 58.8 | 24.5407 | $p < 0.05$ |
| 9 | In video calls, I hear worse than if the other person was in the room with me. | 23 | 39.1 | 17.4 | 43.5 | 34 | 32.4 | 23.5 | 44.1 | 3.9865 | $p < 0.05$ |
| 10 | In video calls, I hear worse than if I was talking to the person on the telephone. | 23 | 43.5 | 26.1 | 30.4 | 34 | 47 | 20.6 | 32.4 | 214.5468 | $p < 0.05$ |
| 11 | I enjoy group video calls (involving more than two people). | 23 | 26.1 | 26.1 | 47.8 | 34 | 35.3 | 23.5 | 41.2 | 18.2354 | $p < 0.05$ |
| 12 | Subtitles on video calls help. | 23 | 21.8 | 21.7 | 56.5 | 34 | 44.1 | 20.6 | 35.3 | 124.3492 | $p < 0.05$ |
| 13 | Since lockdown began, I have been wearing my hearing aids less than usual. | 23 | 52.2 | 8.7 | 39.1 | 34 | 79.4 | 11.8 | 8.8 | 247.4253 | $p < 0.05$ |
| 14 | I am more affected by my hearing loss than usual. | 23 | 21.8 | 21.7 | 56.5 | 34 | 32.3 | 32.4 | 35.3 | 61.8229 | $p < 0.05$ |
| 15 | Televised updates about COVID-19 are easy for me to follow. | 23 | 43.5 | 26.1 | 30.4 | 34 | 20.5 | 32.4 | 47.1 | 116.0672 | $p < 0.05$ |
| 16 | Radio updates about COVID-19 are easy for me to follow. | 23 | 47.8 | 26.1 | 26.1 | 34 | 44.1 | 26.5 | 29.4 | 3.441 | NS |
| 17 | My tinnitus has been worse since lockdown started. | 23 | 56.6 | 21.7 | 21.7 | 34 | 79.4 | 5.9 | 14.7 | 37.5199 | $p < 0.05$ |

*Free Text Comments*

The free text comments highlighted the CI subjects' difficulties in some daily situations in this pandemic period. For example, BCI participant #29 and UCI participant #16 expressed the difficulty of easily living during online meetings: #29 noted: "The subtitles in video calls or webinars are almost never included", as well as #16: "I would like lessons via video call to be subtitled." Concern about the lack of audiology services for people with hearing loss was expressed by different participants: "I suggest an increase of subtitles in places such as train stations and the obligation in the workplace to purchase FM systems to simplify for people with hearing loss listening with masks, especially in noisy places"(participant #12, UCI user)."Make certified transparent masks mandatory in a public place in the presence of deaf people and add Italian subtitles to all video call programs for full inclusion and accessibility"(participant #40, BCI user). "I think that the news with LIS (Italian sign language) translation is useless. I'd like to have subtitles on all tv news programs" (participant #42, UCI user). Emotional reactions were evident in other responses: "I wish people would not be angry if I ask that the mask be pulled down for a moment" (participant #11, BCI user). Participant #33, BCI, reported a frequent technical problem associated with the use of personal protective equipment (PPE) in HA and CI owners: "I often get damaged cables or accessories with the mask on". However, some positive sentiments and hope for the future were also indicated: "I am taking advantage of this period to limit the use of lip-reading, work video calls are a source of pride for me and my suggestion to all people with hearing loss is to take advantage of the difficulties of this period to take the benefits from when everything will get better" (Participant #74, BCI user).

## 4. Discussion

The arrival and rapid spread of the COVID-19 pandemic triggered a global increase in the use of face coverings in an attempt to reduce transmission of the virus [3]. One unintended consequence of face coverings is their impact on communication, especially for people with impairments, such as hearing loss, who are living with an extra condition of isolation besides social distancing [15]. Acoustic degradation of speech created by different face coverings may have a larger negative effect on speech recognition for listeners with hearing loss compared to normal-hearing listeners [10]. Some cultural associations in Italy have denounced the problems that patients suffering from hearing impairments have understanding and communicating with the health personnel constantly wearing PPE, especially when they have to go to the emergency room for any reason. Previous work [16] has demonstrated that listeners with severe-to-profound hearing loss experience significantly poorer speech recognition when the speaker is wearing a conventional or transparent surgical mask, as compared to unmasked conversations. Until now, there have been no studies in the literature that have described the subjective effect of lockdown and therefore of social distancing on CI users in different aspects of life, such as behavior, emotions, practical issues, and hearing performance. From the results of our online survey, the impact the COVID-19 pandemic has had on CI patients has emerged and certain aspects of the situation have proved especially challenging. Wearing face masks, physical and social distancing, e-learning, and virtual communications, have provided important problems for CI users. The impact of the pandemic also revealed an increase in anxiety and difficulty in social interaction. Answers given to questions relating to problems of an emotional and practical nature, revealed that the oldest subjects (over 61 years of age) using CI were more affected by the negative effects of lockdown when they were UCI users rather than BCI users. The oldest UCI users expressed a higher necessity for transparent face masks worn by key workers probably due not only to the increased difficulty in lip-reading but also to the greater difficulty in interpreting the interlocutor's emotions. At the same time, they expressed greater discomfort in attending social gatherings and in speaking to people wearing a face mask or from a distance. The oldest UCI users also seemed to enjoy group video calls less than BCI users, and they seemed to be more affected by their hearing loss

during lockdown than BCI users of the same age. Considering the practical issues theme, probably the lack of access to audiological services has affected a considerable number of the respondents. Previous studies [17–19] demonstrated that in older adults, hearing loss is independently associated with an increased rate of cognitive decline, and has been identified to be a modifiable risk factor for dementia. Sarant et al. (2019) [20], in a 5-year longitudinal study, investigated the impact of CI on cognitive function in older people with severe-profound hearing loss, and whether remediation of hearing loss could delay the onset of cognitive impairment. Significant benefits of CI were reported in terms of speech perception, communication ability, and the quality of life.

The benefits of BCI compared to UCI in adults with postlingual deafness have already been reported in terms of binaural benefit [21,22]. In everyday life, sounds come from different directions and usually in background noise; our study confirms that patients wearing BCI significantly benefit from their second implant in these situations, especially in the case of the social distancing experience and during conversation filtered by masks. Dunn et al. [23] using a cognitive load test which demonstrated that BCI listeners are presumably better at processing speech while attending to other simultaneous tasks compared to UCI listeners, perhaps because they are able to segregate the signal from the background noise.

The results of our study showed that the proportion of BCI user responses significantly differed from UCI responses in all domains, except the practical issues. It seems that BCI users reported lower problems associated with the lockdown compared to UCI users. Using a BCI is not only associated with better auditory performance, especially in noise, but also with the reduced negative effects of social distancing and lockdown in adults with severe and profound deafness. Another practical implication that came out of our survey (especially from the free text comments) was that the adoption of clear face masks and the use of live subtitles on video calls may alleviate some of the difficulties and anxieties that the population with hearing loss experience, as was evident in the study by Naylor et al. [13].

## 5. Limitations of the Study

The study has some limitations that are worth mentioning.

Firstly, participation was voluntary, which is a potential source of selection bias.

The information in our possession on gender, age, and possession of CI (UCI or BCI) are self-reported data that cannot be independently verified (potential sources of bias). Furthermore, there are no best practice guidelines to conduct qualitative studies using data from Facebook (our methodology can be considered self-identification research).

## 6. Conclusions

The aim of the study was to ascertain the perceived effects of social restrictions during the COVID-19 lockdown on CI users. From our results, it emerged that CI users experienced the lockdown negatively as regards behavior, emotions, hearing performance, and practical issues. CI subjects with UCI in old age suffered more from the experience of lockdown than subjects with BCI, given the impossibility of being able to exploit the effect of binaural in difficult conditions responsible for an increased cognitive load. Finally, women using CI reported more practical problems during the lockdown than men. Results similar to those of Naylor's study were obtained from the free text comments of the survey: key workers should be provided with transparent face masks; patients should be informed about the availability of live subtitling on video-calling platforms; and device signal processing modes and accessories compatible with video-calling should be developed and propagated.

**Author Contributions:** Conceptualization, G.C., A.M. and N.Q.; methodology, R.G., A.M. and N.Q.; formal analysis, G.C. and R.G.; investigation, G.C., A.M. and N.Q.; data curation, G.C. and A.M.; writing—original draft preparation, G.C.; writing—review and editing, G.C., A.M., E.N. and N.Q.; supervision, A.M. and N.Q. All authors have read and agreed to the published version of the manuscript.

**Funding:** This research received no external funding.

**Institutional Review Board Statement:** The study was approved by the local hospital Ethics Committee (Comitato Etico Indipendente Azienda Ospedaliero-Universitaria "Consorziale Policlinico Bari") (Prot. n° 0019580/28/02/2022).

**Informed Consent Statement:** Informed consent was obtained from all subjects involved in the study.

**Data Availability Statement:** Not applicable.

**Acknowledgments:** Not applicable.

**Conflicts of Interest:** The authors declare no conflict of interest.

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
