# Peer review of "The Impact of the COVID-19 Lockdown on Quality of Life in Adult Cochlear Implant Users: A Survey Study"

_audiolres, doi:10.3390/audiolres12050052_

Round 1
Reviewer 1 Report
Please correct/explain:
1. Abstract line 19: add - subject with BCI in the same age regarding...
2. In table 1 you described the whole group and mentioned about Female UCI and BCI. Was the male UCI/BCI percentage ommited intentionally?
3. In your questionaire there were 5 categories, then you reduced them to 3. Why?
4. In line 122-123 you mentioned that "...lock-down ... neither increased the tinnitus." In the same article several times you mentioned that it did (line 244 and others). Why such an inconsistency?
5. Why in questionaire for point 12 (subtitles help) the mean result was neutral (3) but in free comments as you mentioned the need of them was clearly visible? Patients didn't understand this quesion?
Reviewer 2 Report
This work is interesting nowadays, but the number of participants is insignificant. The title of this manuscript should change since all participants are adults and "baby boomers". So, the title is not including infants and children. There are some in-text citations where authors have to change the form of these from [9][10] to [9,10]. You should change the term "sex" to "gender" since the first term is obsolete. Finally, the way the authors present their statistical analysis needs some improvement since p values are not mentioned following a reference system. There are lines where the authors mention (p=..) and other places (p = ...) (see lines 130, 135, 136). The term "PPE" is exclaimed mistakenly in the section of discussion (line 192) while they mentioned it for the first time in line 175. Please do not add new terms in the discussion section. See line 180 and the full stops you have used. "women using CI reported more practical problems during the lockdown," but women were 72% of this sample!
Round 2
Reviewer 2 Report
Dear authors
1) There are no best practice guidelines to help you conduct qualitative studies using data from Facebook. Your methodology is a research self-identification since you have recruited participants to answer this questionnaire, and you must mention this kind of methodology.
2) you have conducted your research based on a Facebook group with almost 15K members. How do you know you have found the necessary participants wearing CI? How do you know their age, gender? How do you know that they are real profiles? The only question to enter this group is "Are you opposed to the right of choice of a parent who chooses a cochlear implant to give their profoundly deaf child a chance to hear?"
3) what are the exclusion criteria have you used since there is no in-person contact with participants to reduce misrepresentation of participants and potentially counterfeit responders
Round 3
Reviewer 2 Report
The methodology you have used can not be accepted, and the outcome is undermined accordingly.
Author Response
We’d like to thank Reviewer 2 for the comments on my work but we disagree because we think that our methodology follows that of the study by Naylor et all (Naylor G, Burke LA, Holman JA. Covid-19 Lockdown Affects Hearing Disability and Handicap in Diverse Ways: A Rapid Online Survey Study. Ear Hear. 2020 Nov/Dec;41(6):1442-1449. doi: 10.1097/AUD.0000000000000948.)